# The effects of sociodemographic factors on help-seeking for depression: Based on the 2017–2020 Korean Community Health Survey

JinYoung Lee[1], Kyeong-Sook Choi[2], Ji-Ae Yun[2]*

1 Department of Neuropsychiatry, Eulji University School of Medicine, Daejeon, Republic of Korea,
2 Department of Neuropsychiatry, Daejeon Eulji Medical Center, Eulji University School of Medicine, Daejeon, Republic of Korea

* jayun5959@gmail.com, jayun5959@eulji.ac.kr

**Data Availability Statement:** Korea Community Health Survey database are available from the Korea Disease Control and Prevention Agency.

## Abstract

When individuals face psychological difficulties that exceed their resources, consulting professionals for mental health treatment can be an effective way to overcome these difficulties. However, in general, only a few patients receive treatment for depression. The goal of the present study was to explore the help-seeking behaviors of currently employed individuals with depression and the factors influencing their help-seeking behaviors. This study used raw data from the Korean Community Health Survey (KCHS) obtained from 2017 to 2020. A total of 6,505 employed individuals, who responded as having experiences of sadness or hopelessness that caused problems in their lives for more than two weeks and who scored more than nine points on the Patient Health Questionnaire-9, were included in our analysis. Help-seeking behavior was measured as receiving expert advice due to feelings of sadness or hopelessness. Of the 6,505 people with depression, only 1,781 (27.38%) received professional counseling for it. Male participants (adjusted odds ratio [aOR] = 1.31, 95% confidence interval [CI] = 1.157–1.487), those aged 45–64 years (aOR = 1.192, 95% CI = 1.022–1.389) and more than 75 years (aOR = 1.446, 95% CI = 1.059–1.973), those not having a Medical Aid program (aOR = 1.750, 95% CI = 1.375–2.226), and those having low educational levels (aOR = .896, 95% CI = .830–.968) were less likely to seek professional help for depression. Our study found that help-seeking behaviors for depression in the Korean population were low. Furthermore, we identified the characteristics associated with individuals with depressive symptoms who chose not to receive help from mental health professionals. The results of this study provide insights to guide national interventions to increase help-seeking behaviors for depression.

## Introduction

Depression is a common mental disorder, and the total number of people living with depression worldwide is approximately 322 million [1]. The lifetime prevalence of major depressive disorder in Korean adults is 5.6%, which implies that one in 20 people experience a depressive episode at some point in their lifetime [2]. It is estimated that depression will be the leading

Researchers interested in the data can request access by sending a proposal to the data access committee at the following link: https://chs.kdca.go.kr/chs/rawDta/rawDtaPrncplMain.do These data are available free of charge. The authors did not have special access privileges to these data sets.

**Funding:** This research was supported by Research of Korea Disease Control and Prevention Agency (2021-11-020). The funders had no role in study design, data collection and analysis, decision to publish, or preparation of the manuscript. There was no additional external funding received for this study.

**Competing interests:** The authors have declared that no competing interests exist.

cause of disease burden worldwide by 2030 [3], as depression is related to high mortality and morbidity. Males and females with depression are 20.9 and 27 times more likely to commit suicide, respectively, than the general population [4]. Recent studies have shown that depression is a risk factor for cardiovascular death [5–7]. In addition to mortality, functional impairment and disability associated with depression are significantly higher [8,9].

When people have to deal with problems or events that they are unable to cope with by themselves, it may be helpful to reach out to mental health professionals for receiving appropriate support, advice, or assistance. Help-seeking from professionals refers to these communications [10]. Help-seeking can be measured in two ways: in the form of attitudes, intentions, and actions to seek help; and by any factors related to each step in seeking help, from initial access to ongoing service use [11].

However, people often avoid or delay seeking professional help for depression. Although many existing studies have reported varying results due to different methodologies, the proportion of help-seeking behaviors among people with depression has been reported to range from 13 to 50% [12,13]. Previous studies have shown that, in many cases, individuals avoid or delay treatment due to the cost of medical care [14]. Cognitive behavioral therapy (CBT) is known to be effective for depression, but considering that it usually requires at least 10 sessions and that most psychotherapy fees are charged per session, the cost aspect is a major obstacle to treatment seeking [15].

In the case of South Korea, however, the cost barrier for medical care is relatively low owing to the National Health Insurance, which provides universal medical coverage for the entire population. As an example in this regard, the number of outpatient visits per capita in Korea (14.7 visits per year vs. the Organization for Economic Cooperation and Development [OECD] average: 5.9 visits per capita per year on average) was the highest among OECD countries [16]. In light of this, help-seeking behavior toward mental health professionals in Korea should consider a number of complex factors beyond the cost.

Though many studies have investigated help-seeking for depression, most have focused on groups vulnerable to depression such as rural residents, older adults, and unemployed people [17–21]. Since various sociodemographic factors are related to help-seeking behavior and depression, it is difficult to generalize results of these studies to all population groups in the community. For the working population, even minor levels of depression are associated with a reduction in occupational function and productivity loss [22], leading to greater social burden [23,24]. It might be pivotal to identify the factors that hinder their help-seeking behaviors. Moreover, to the best of our knowledge, no studies have been conducted on large-scale samples in Korea.

Therefore, the goal of the present study was to explore the help-seeking behaviors of people with depression, who are currently employed, and the factors influencing their help-seeking behaviors using the Korean Community Health Survey (KCHS). Through this study, we intend to provide basic data to promote mental health and encourage help-seeking behaviors by those suffering from depression.

## Methods

### Data and subjects

This study used the KCHS raw data obtained from 2017 to 2020. The KCHS is a nationwide health interview survey conducted by the Korea Disease Control and Prevention Agency (KDCA) annually to estimate disease prevalence, morbidity, personal lifestyle, and health behaviors. It provides local autonomous entity data that can be used to plan, implement, monitor, and evaluate community health promotion and disease prevention

programs. The survey gathered information about households, health behaviors, health problems, medical use, accidents, quality of life, and socioeconomic status through face-to-face interactions between trained interviewers and the respondents. The target population of KCHS were adults aged $\geq$ 19 years and those living in the jurisdiction of one community health center [25]. The study design and data analysis protocol were reviewed and approved by the Institutional Review Board of Eulji University Hospital (EMC 2022-02-020). Consent to participate was not applicable since the KDCA hosts the KCHS raw data in a public website.

A total of 915,089 participants completed the survey over four years (2017–2020). Since we aimed to investigate help-seeking behaviors for depression in currently employed adults, unemployed participants were excluded from this study. Among those employed (n = 561,654), we selected participants who mentioned experiencing feelings of sadness or hopelessness that negatively affected their lives for more than two weeks (4.59%, n = 25,790), and those who scored more than nine points on the Patient Health Questionnaire-9 (PHQ-9) (2.92%, n = 16,390). Ultimately, 6,505 (1.16%) individuals who met both conditions were included in the analysis.

## Measures

The KCHS is conducted by a trained interviewer who visits the selected sample households and conducts one-on-one interviews with all participants using an indirect entry method. The interview takes approximately 20–30 min per person [25].

The item that investigated help-seeking of the participants from professionals, had a response option of yes/no depending on whether they had ever received expert advice due to feelings of sadness or hopelessness: "During the past year, have you felt sad or hopeless enough to interfere with your daily life for more than two weeks in a row? If so, have you ever received expert advice (medical institution, professional counseling institution, public health center, etc.) because of these problems?" Sociodemographic factors included gender, age, housing type, residential region (dong meaning urban/eup-myeon meaning rural), and type of National Health Insurance (Medical Aid program (MA); MA is a public assistance scheme that secures the minimum livelihood of low-income households and assists with self-help by providing medical services/ National Health Insurance (NHI); NHI covers the whole population as a compulsory scheme [26]), monthly household income, economic activity, occupation, educational level, and marital status.

Participants were categorized in terms of their age as follows: 19–44 years, 45–64 years, 65–74 years, and 75 years or more [27]. Monthly household income was classified into eight groups: 1) 0.5 million Korean won (KRW) or less; 2) 0.5–1 million KRW; 3) 1–2 million KRW; 4) 2–3 million KRW; 5) 3–4 million KRW; 6) 4–5 million KRW; 7) 5–6 million KRW; and 8) 6 million or more. Occupations were classified into four groups: 1) managers/professionals; 2) clerks/service and sales workers; 3) agricultural, forestry, and fishery workers; and 4) mechanical and manual laborers. Educational level was classified into five groups: 1) elementary school or less, 2) middle school, 3) high school, 4) college, and 5) graduate school or higher. Marital status was classified into three groups: 1) spouse, 2) separated/divorced/widowed, and 3) never married.

The Korean version of the PHQ-9, using the criteria from the Diagnostic and Statistical Manual of Mental Disorders, Fourth Edition (DSM-IV) for major depressive disorder, was used for the screening of depression. The PHQ-9 has a range of 0–27 points and asks how often an individual has experienced depressive symptoms in the nine items over the past two weeks. An et al. suggested that the optimal cutting point was calculated as 9 points with 88.5%

sensitivity and 94.7% specificity [28]. According to Lim et al., both 9 and 10 points were found to have 88% sensitivity, but the specificity of 9 points was 86% and that of 10 points was 52% [29]. Thus, 9 points were used as a cut-off for screening depression [2,28,29]. Cronbach's alpha for the PHQ-9 was 0.802 in the present study. Subjective stress was measured on a four-point Likert scale (very much/much/a little/rarely) according to how much stress the individual felt in their daily life. To perform the analyses, we classified the response regarding stress into two groups: 1) very much/much, and 2) little/rarely.

## Statistical analysis

Frequency analysis, chi-square test, t-test, logistic regression analysis, and factor analysis were performed using SPSS v25.0 for the data analysis. Chi-square tests and t-tests were used to determine the significance of the differences between participants who did and did not seek help. Multiple logistic regression was used to obtain adjusted odds ratios (ORs) and 95% confidence intervals (95% CI). Help-seeking behavior was coded as 0, which stands for help-seeking behavior and 1, stands for no help-seeking behavior. Subjective stress and severity of depressive symptoms (i.e., PHQ-9 score) were included in the multiple logistic analysis as control variables because the relationship between help-seeking and these variables has been demonstrated by prior research [30,31]. A previous study (Jeon et al., 2020) using the BDI-II (Beck Depression Inventory) suggested that there were two latent factors of depression [32]; moreover, we expected that the PHQ-9 would have two factors. Thus, we conducted a factor analysis on the PHQ-9 to explore which depressive symptoms had a greater effect on help-seeking. For the factor analysis of the PHQ-9, data from 915,089 participants, including those who were not depressed, were used to identify sub-factors for depression in the general population. Since we expected a significant correlation between the sub-factors of PHQ-9, principal component analysis with direct oblimin rotation ($\delta = 0$) was used to extract the main factors with eigenvalues >1.0.

## Results

### Sample characteristics

The general characteristics of the study population and their help-seeking behaviors are summarized in Table 1. Depression was experienced by 5.0–5.8% of the total participants from 2017 to 2020 [33] and 4.59% of the sample included in our analysis. Of the 6,505 people with depression, only 1,781 (27.38%) had received expert advice for depression. Compared to participants who sought help (n = 1,781), those who did not (n = 4,724) were older ($\chi 2$ = 22.048, p < .001), had lower levels of education ($\chi 2$ = 17.377, p < .001), were mostly males ($\chi 2$ = 11.769, p < .001), had NHI ($\chi 2$ = 20.753, p < .001), and were agricultural, forestry, fishery, mechanical, and manual laborers ($\chi 2$ = 8.349, p < .05).

The factor analyses of the PHQ-9 in the sample are summarized in Table 2. The Kaiser–Meyer–Olkin (KMO) value was .874, and Bartlett's test of sphericity produced an approximation of $\chi^2$ = 2374168.654 (df = 36, p < 0.001), indicating that sampling adequacy for factor analysis was good. As expected, two factors of depression were extracted from the analysis: one was related to somatic-affective symptoms (items 1–5), and the other was a cognitive factor related to depression (items 6–9). This explained 55.8% of the variance.

Monthly household income, economic activity, and marital status were used as confounding variables in the logistic regression analysis. The aORs and 95% CIs for help-seeking by sociodemographic factors, subjective stress, and severity of depressive symptoms are shown in Table 3. Multiple logistic regression models revealed that sex, age, type of national health insurance, and educational level influenced help-seeking for depression. It was found

**Table 1. General characteristics of the study population who was screened as having depression (N = 6,505).**

| Categories | N = 6,505 | Help-seeking[a] | | | | χ2 / t |
| | | Yes | | No | | |
| | | n = 1,781 | 27.4% | n = 4,724 | 72.6% | |
|---|---|---|---|---|---|---|
| **Gender** | | | | | | |
| Male | 2,301 | 571 | 24.80% | 1,730 | 75.20% | 11.769** |
| Female | 4,204 | 1,210 | 28.80% | 2,994 | 71.20% | |
| **Age group (in years)** | | | | | | |
| 19–44 | 2,884 | 864 | 30% | 2,020 | 70% | 22.048*** |
| 45–64 | 2,395 | 631 | 26.30% | 1,764 | 73.70% | |
| 65–74 | 765 | 186 | 24.30% | 579 | 75.70% | |
| 75 or more | 461 | 100 | 21.70% | 361 | 78.30% | |
| **Housing type** | | | | | | |
| General house | 4,029 | 1,078 | 26.80% | 2,951 | 73.20% | 2.066 |
| Apartment housing | 2,476 | 703 | 28.40% | 1,773 | 71.60% | |
| **Residential region** | | | | | | |
| Dong (urban) | 3,955 | 1,088 | 27.50% | 2,867 | 72.50% | .086 |
| Eup-myeon (rural) | 2,550 | 693 | 27.20% | 1,857 | 72.80% | |
| **Type of National Health Insurance** | | | | | | |
| Medical Aid Program | 332 | 127 | 38.30% | 205 | 61.70% | 20.753*** |
| National Health Insurance | 6,170 | 1,654 | 26.80% | 4,516 | 73.20% | |
| **Monthly household income** | | | | | | |
| 50 or less | 332 | 77 | 23.20% | 255 | 76.80% | 13.43 |
| 50–100 | 776 | 204 | 26.30% | 572 | 73.70% | |
| 100–200 | 1,174 | 355 | 30.20% | 819 | 69.80% | |
| 200–300 | 1,134 | 290 | 25.60% | 844 | 74.40% | |
| 300–400 | 897 | 245 | 27.30% | 652 | 72.70% | |
| 400–500 | 646 | 189 | 29.30% | 457 | 70.70% | |
| 500–600 | 581 | 149 | 25.60% | 432 | 74.40% | |
| 600 or more | 853 | 249 | 29.20% | 604 | 70.80% | |
| **Economic activity** | | | | | | |
| Yes | 6,480 | 1,773 | 27.40% | 4,707 | 72.60% | .109 |
| No | 23 | 7 | 30.40% | 16 | 69.60% | |
| **Occupation** | | | | | | |
| Managers/professionals | 1,885 | 546 | 29% | 1,339 | 71% | 8.349* |
| Clerks/service/sales | 1,726 | 494 | 28.60% | 1,232 | 71.40% | |
| Agricultural, forestry, fishery | 958 | 243 | 25.40% | 715 | 74.60% | |
| Mechanical/manual laborers | 1,936 | 498 | 25.70% | 1,438 | 74.30% | |
| **Educational level** | | | | | | |
| Elementary school | 1,326 | 325 | 24.50% | 1,001 | 75.50% | 17.377** |
| Middle school | 643 | 160 | 24.90% | 483 | 75.10% | |
| High school | 1,997 | 541 | 27.10% | 1,456 | 72.90% | |
| College | 2,312 | 679 | 29.40% | 1,633 | 70.60% | |
| Graduate school | 219 | 75 | 34.20% | 144 | 65.80% | |
| **Marital status** | | | | | | |
| Spouse | 3,387 | 907 | 26.80% | 2,480 | 73.20% | 2.186 |
| Separated/divorced/widowed | 1,440 | 394 | 27.40% | 1,046 | 72.60% | |
| Never married | 1,666 | 479 | 28.80% | 1,187 | 71.20% | |
| **Subjective stress** | | | | | | |

*(Continued)*

**Table 1.**  (Continued）

| Categories | N = 6,505 | Help-seeking[a] | | | | χ2 / t |
| | | Yes | | No | | |
| | | n = 1,781 | 27.4% | n = 4,724 | 72.6% | |
| Very much/much | 5,501 | 1,553 | 28.20% | 3,948 | 71.80% | 12.784*** |
| A little/rarely | 1,002 | 228 | 22.80% | 774 | 77.20% | |
| **PHQ-9** | | | | | | |
| Somatic-affective symptoms | | 9.92 | | 9.53 | | 4.79*** |
| Cognitive symptoms | | 3.91 | | 3.16 | | 9.90*** |

Notes: PHQ-9 = Patient Health Questionnaire 9 items, Somatic-affective symptoms = the sum of PHQ-9 items 1–5 (loss of interest, depressive mood, disturbed sleep, feelings of tiredness, disturbed appetite), PHQ-9 Cognitive symptoms = the sum of PHQ-9 items 6–9 (feelings of guilt, poor concentration, psychomotor agitation or retardation, suicide).

[a] Help-seeking from professionals measured as yes/no, in terms of whether they had ever received expert advice due to feelings of sadness or hopelessness.

*p< .05

**p< .01

***p< .001.

that males were less likely to seek help for depression than females (adjusted odds ratio (aOR) = 1.31; 95% CI = 1.157–1.487). Individuals aged 45–64 years and 75 years or more were less likely to seek help for depression than those aged 19–44 years (45–64 aOR = 1.19; 95% CI = 1.022–1.389; 75 or more aOR = 1.45; 95% CI = 1.059–1.973). Individuals who did not have MA were less likely to seek help for depression than individuals with the program (aOR = 1.75; 95% CI = 1.375–2.226). Low educational levels were associated with significantly lower help-seeking (aOR = .90; 95% CI = .820–.968). Additionally, the more severe the depressive symptoms, the higher the help-seeking behavior. Depressive symptoms related to guilt, concentration, psychomotor changes, and suicide had a greater influence on help-seeking behavior than depressive symptoms related to loss of interest, mood, sleep, tiredness, and appetite (somatic-affective symptoms aOR = .97; 95% CI = .950–.989; cognitive symptoms aOR = .90; 95% CI = .882–.921).

**Table 2.  The factor analyses of the PHQ-9 in the sample at study entry (N = 915,089).**

| Item | Pattern coefficient | | Structure coefficient | | Communality |
| | 1 | 2 | 1 | 2 | |
| Feeling tired or having little energy | **.847** | .117 | **.791** | -.286 | .637 |
| Trouble falling or staying asleep, or sleeping too much | **.756** | .068 | **.724** | -.292 | .528 |
| Poor appetite or overeating | **.672** | -.064 | **.718** | -.569 | .497 |
| Little interest or pleasure in doing things | **.622** | -.124 | **.702** | -.384 | .477 |
| Feeling down, depressed, or hopeless | **.578** | -.294 | **.682** | -.421 | .582 |
| Moving or speaking so slowly that other people could have noticed? Or the opposite—being so fidgety or restless that you have been moving around a lot more than usual | -.115 | **-.822** | .418 | **-.803** | .598 |
| Thoughts that you would be better off dead or of hurting yourself in some way | .046 | **-.781** | .276 | **-.767** | .646 |
| Feeling bad about yourself—or that you are a failure or have let yourself or your family down | .100 | **-.717** | .442 | **-.765** | .593 |
| Trouble concentrating on things, such as reading the newspaper or watching television | .055 | **-.656** | .367 | **-.682** | .468 |

Note: Major loadings for each item are shown in boldface.

**Table 3. Results of multiple logistic regression on help-seeking for depression.**

| Variables | | aOR | 95% CI |
|---|---|---|---|
| **Gender** | Female | | |
| | Male | 1.311*** | (1.157–1.487) |
| **Age group (in years)** | 19–44 | | |
| | 45–64 | 1.192* | (1.022–1.389) |
| | 65–74 | 1.218 | (0.945–1.568) |
| | 75 or more | 1.446* | (1.059–1.973) |
| **Residential region** | eup-myeon (rural) | | |
| | dong (urban) | 1.073 | (0.939–1.225) |
| **Type of National Health Insurance** | Medical Aid program | | |
| | National Health Insurance | 1.750*** | (1.375–2.226) |
| **Monthly household income** | | 0.997 | (0.981–1.013) |
| **Occupation** | managers/professionals | | |
| | clerks/service/sales | 0.955 | (0.817–1.116) |
| | agricultural, forestry, fishery | 0.907 | (0.712–1.156) |
| | mechanical/manual laborers | 0.997 | (0.837–1.188) |
| **Educational level** | | 0.896** | (0.830–0.968) |
| **Marital status** | never married | | |
| | spouse | 0.895 | (0.767–1.043) |
| | separated/divorced/widowed | 0.877 | (0.725–1.060) |
| **Subjective stress** | very much/much | | |
| | a little/rarely | 1.192* | (1.011–1.406) |
| **PHQ-9 factor1** | somatic-affective symptoms | 0.969** | (0.950–0.989) |
| **PHQ-9 factor2** | cognitive symptoms | 0.901*** | (0.882–0.921) |

Notes: Help-seeking coding: 0 = yes; 1 = no; aOR = adjusted odds ratio, CI = confidence interval, PHQ-9 factor1 = the sum of PHQ-9 items 1–5 (loss of interest, depressive mood, disturbed sleep, feelings of tiredness, disturbed appetite), PHQ-9 factor 2 = the sum of PHQ-9 items 6–9 (feelings of guilt, poor concentration, psychomotor agitation or retardation, suicide). As control variables, subjective stress and PHQ-9 sub-factors were included in this analysis.

*p< .05

**p< .01

***p< .001.

## Discussion

This study was conducted to identify the factors related to help-seeking behavior among currently employed adults in Korea. Based on KCHS data accumulated from 2017 to 2020, 5–5.8% of these adults were found to experience significant depressive symptoms. Our findings demonstrated that 27.38% of employees with depression did not seek help. Furthermore, it was found that being male, being aged 45–64 years and 75 years or older, having NHI, and having low education levels were related to less professional help-seeking behavior.

One possible reason for this is stigma and a negative attitude toward mental disorders. Stigma refers to the negative effects of a label placed on any group, such as a racial minority or those who have mental illness [34]. Perceived public stigma and anticipated self-stigma may lead individuals to avoid seeking help, especially in the case of depression [35]. Many people have reported feeling embarrassed about seeking help from professionals, believing that other people would judge them negatively [12]. Additionally, self-stigmatization such as personal discriminatory attitudes significantly decreased willingness to seek psychiatric help [36]. For example, in the case of males, masculine social norms such as self-reliance and emotional

control can make it difficult for males to seek help when they acknowledge their depression [37]. Korean studies also found that as men's gender roles were emphasized, depression in them increased, which had a negative effect on help-seeking behavior among them [38,39]. Contrarily, females tend to have positive attitudes toward psychological openness and thus exhibit more favorable mental health services than males [40]. In our study, the degree of seeking help for older adults was generally lower than that for young adults, especially in groups aged 45–64 years and 75 years and older. This may also be because older people have a negative attitude toward mental disorders [41]. In the case of older adults, concerns regarding cost, transportation, inconvenience, and confidence can also be barriers to mental health help-seeking [42].

Being older, being male, and having a lower educational status are consistent findings throughout previous studies that impede help-seeking behavior in depressed individuals [43]. The result of no significant association between income and help-seeking behavior is also consistent with previous studies. Authors reported that income as a single indicator may not be sensitive enough to detect socioeconomic differences in medical service use [43].

Interestingly, it was found that individuals with an NHI compared to those with MA seek help less frequently. One possible reason is that MA type I beneficiaries are not required to provide copayments for any medical utilization or, in the case of those with type II beneficiaries, have minimum copayment rates of up to 15% [44]. There is evidence that medical coverage of MA increased the number of outpatient visits [44]. However, as we mentioned earlier, our results cannot be interpreted simply as lowering help-seeking behavior in individuals with NHI due to an increase in out-of-pocket spending. This is because the total number of outpatient visits in Korea is higher than in other countries [16]; however, the mental health service utilization rate in Korea is much lower at 7.2% compared to other countries (Canada 46.5%, United States 43.1%) [45].

The practical implication that can be considered based on these results is that it is possible to consider ways to further expand access to mental health services for currently employed groups of males, older adults, low-educated groups, and NHI groups. According to the Commission on Social Determinants of Health conceptual framework [46], socioeconomic status acts as a structural determinant, which impacts health inequity. Identifying socioeconomic factors related to help-seeking behavior can generate useful data for intersectoral policy approaches and social capital to complement individual socioeconomic vulnerability. Thus, public policies should tackle these populations individually using tailored strategic approaches to improve their help-seeking behavior.

Our study has some limitations. First, the cross-sectional design precludes causal inferences. Therefore, the results should be interpreted cautiously. Future studies with longitudinal designs would provide more robust evidence of causality. Second, this study targeted employed individuals alone; therefore, bias cannot be ruled out, and the results of this study cannot represent the entire community population. Third, this study did not consider all factors that may affect help-seeking behavior. Although possible confounding factors were added to the multivariate logistic regression analysis to overcome these limitations, more factors should be considered in future studies. Further, although a trained interviewer conducted the interview to measure depression and subjective stress, the participants were not clinically diagnosed with depression by a mental health professional. Moreover, subjective stress was measured by one simple question rather than a validated tool. Furthermore, since data were acquired via recall during an interview, recall bias cannot be excluded. Finally, there is a possibility that participants underreported their level of depression and help-seeking behavior due to the stigma of mental illness [47]. Nevertheless, this study is meaningful as it is the first Korean study to identify the current status of help-seeking behavior in currently employed

adults and to identify the influence of sociodemographic factors using KCHS data. This study provides more generalized information on help-seeking using the community health survey data obtained using a standardized method. These results provide basic data for mental health promotion through improvements in help-seeking behaviors.

## Author Contributions

**Conceptualization:** Kyeong-Sook Choi.

**Formal analysis:** JinYoung Lee.

**Funding acquisition:** Kyeong-Sook Choi.

**Supervision:** Kyeong-Sook Choi, Ji-Ae Yun.

**Writing – original draft:** JinYoung Lee.

**Writing – review & editing:** Ji-Ae Yun.

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
