## [Decision Letter · Decision Letter 0]

30 Aug 2022

PONE-D-22-18284The Effects of Sociodemographic Factors on Help-Seeking for Depression Based on the 2017-2020 Korean Community Health SurveyPLOS ONE

Dear Dr. Yun,

Thank you for submitting your manuscript to PLOS ONE. After careful consideration, we feel that it has merit but does not fully meet PLOS ONE’s publication criteria as it currently stands. Therefore, we invite you to submit a revised version of the manuscript that addresses the points raised during the review process.

Both reviewers indicate substantial reporting issues that need clarification before considering publication of the manuscript.

We look forward to receiving your revised manuscript.

Kind regards,

Pedro Vieira da Silva Magalhaes, M.D., Ph.D.

Academic Editor

PLOS ONE

Journal Requirements:

"This research was supported by Research of Korea Disease Control and Prevention Agency (2021-11-020)."

"This research was supported by Research of Korea Disease Control and Prevention Agency (2021-11-020)."

Reviewers' comments:

Reviewer's Responses to Questions

**Comments to the Author**

1. Is the manuscript technically sound, and do the data support the conclusions?

Reviewer #1: Partly

Reviewer #2: Partly

2. Has the statistical analysis been performed appropriately and rigorously? 

Reviewer #1: Yes

Reviewer #2: Yes

3. Have the authors made all data underlying the findings in their manuscript fully available?

Reviewer #1: Yes

Reviewer #2: Yes

4. Is the manuscript presented in an intelligible fashion and written in standard English?

Reviewer #1: Yes

Reviewer #2: Yes

5. Review Comments to the Author

Reviewer #1: Review comments on the manuscript PONE-D-22-18284

Dear authors,

Thank you for your research article and your work in this important scientific field. The article is very concise (maybe too conscience). Some changes should be made to improve readability and scientific transparency before publication.

Yours sincerely and all the best,

Your anonymous reviewer

Attached as a pdf

Reviewer #2: Thank you for submitting your manuscript. It was quite interesting to read.

Comments

Background

• Please provide information on help for depression available in Korea. How easy or costly is it to seek help for depression?

Methods

• How many people participated in KCHS in total – what percentage makes 6505?

• Please present the inclusion and exclusion criteria more clearly. Why were the exclusion criteria chosen?

• What have been the prevalence for depression in Korea in the time 2017 to 2020? Are the people identified in your investigated sample representative for the prevalence?

• Was help seeking assessed by self-report on one item, multiple items or validated scales? Please specify the assessment. (ll 111-113)

Results

• Ll151 -154: Please specify which populations you investigated. There is obviously the total population of: 915,089, then there is people with depressive symptoms: 9505, then there is people seeking help in the total population 561,654 .. It is quite confusing with all the different populations to understand, which populations your statistics were based on. Please find a way to make it easier to follow. Please also present the relative frequencies in regard to the total population of 915,089.

• Ll 154: I understand that 561,654 people sought help – but then of the depressed people only 1781 were seeking help. I thought, you filtered for help seeking in the beginning? What is the difference between help seeking and professional counseling? Please also provide these data in table 1.

• What population was the factor analysis based on? Please provide N?

• Please explain what aOR is? Please incorporate that abbreviation in the text.

• In table 3 you present OR – or is this also aOR?

Discussion

• Please discuss the following limitations: PHQ-9 just a screener for depressive symptoms, there was no psychological interview validating a depression diagnosis. Please also discuss potential biases of non-reporting for help seeking or recall bias.

Tables

• Please provide a legend below the table 1 explaining abbreviations and asterisks

6. PLOS authors have the option to publish the peer review history of their article (what does this mean?). If published, this will include your full peer review and any attached files.

Reviewer #1: No

Reviewer #2: No

---

## [Author Response · Author response to Decision Letter 0]

25 Oct 2022

ANSWERS TO REVIEWERS

We sincerely thank the reviewers for their interest in our work and for their helpful comments that have helped to greatly improve the manuscript. We tried to do our best to respond to the points raised by the reviewers and address every issue. As indicated below, we have checked all the general and specific comments provided by the reviewers and have made necessary changes according to their suggestions.

Reviewer #1

1. Corresponding mail: If possible a more professional mail address than jayun5959@gmail.com

- Response 1: Thank you for your comment. We have now added one more email address. (Line 17)

2. The term “who did not receive national basic security” is no understandable for me (as a non-Korean reader?). Clarify please 

- Response 2: Thank you for your comment. The terminology of the Korean National Health Insurance system has been modified so that it becomes easily understandable to the non-Koreans. Type of National Health Insurance – Medical Aid/ National Health Insurance (Line 132-136).

3. were more likely not to → confusing statement. Better: were less likely to …

- Response 3: Thank you. We have now revised it according to your comment. (marked in red)

4. with depressed individuals → stigmatising. Please use “people first” language in the whole article (e.g., individuals with depressive symptoms)

- Response 4: Thank you for your comment. We have now revised it according to your comment. (marked in red).

5. Keyword: “psychiatric treatment” doesn’t fit in my opinion, because …”Help-seeking behavior was measured as receiving counseling from professionals (medical institutions, professional counseling institutions, public health centers, etc.)” → professional treatment might be better

- Response 5: Thank you. We have now revised it as per your suggestion. (Line 49).

6. People with depression tend not to seek help because they believe they should solve depressive symptoms on their own without external help → This doesn’t seem a fitting citation because (16) is for youth and adolescents, however here the article authors generalise it for “people”. Furthermore, “wanting to solve issues by oneself” is not something that is so bad, yet here the authors state it in the context with stigmatising attitudes. Why? 

- Response 6: Thank you for your insightful comment. As you pointed out, the reference we cited was judged to be inappropriate. So we decided to delete the sentence. Instead, we have presented a more appropriate evidence that stigma inhibits help-seeking behavior in people. Thus, in the process of revising the manuscript, these contents were transferred to discussion section. (Line 224-230)

7. Especially, depression in the productive population causes more social burden → no citation? This seems necessary to me for such a statement.

- Response 7: Thank you for your recommendation. We agree with you that a citation is needed for the mentioned section. We have now added a relevant citation. (Line 89)

8. Concerning Stigma: Too short paragraph with little reflection of different types of stigmatising attitudes and how they might impact the help-seeking process of people with depressive symptoms. I suggest that the authors reflect a little more on stigma. Such as “internalised”, “anticipated” and “help-seeking” sigma are possible search word I suggest looking up in this context. 

- Response 8: Thank you for your suggestion. As your opinion, our study did not investigate stigma. Therefore, we decided to mention it only in the discussion. (Line 224-230)

- We were able to explain the stigma in more depth and presented the evidences related to help-seeking behavior. (Line 224-230)

9. using the standardized method of the Korean Community Health Survey → confusing sentence, since the Survey is not a standardised method…? Is it a survey that is repeatedly issued ever so often? Clarify please

- Response 9: Thank you for your comment. We have now revised it according to your comment; ‘the standardized method of’ was deleted. (Line 94)

- This survey is conducted annually. This fact has been we added Methods-Data and subjects section. (Line 102)

10. Patients with unemployment were excluded from the study. → Why? These are an especially vulnerable group (also for depression) and concerning stigma/help-seeking and financial barriers you wrote about above, they seem to be of special interest? Please clarify the rationale. 

- Response 10: Thank you for your comment. Although many studies about help-seeking behavior that have been conducted focused on the vulnerable groups, there has been no large-scale Korean study on adults in the community with who are currently employed. Since various socioeconomic factors are related to help-seeking behavior and depression, it is difficult to apply the research results targeting the vulnerable group to them. Therefore, in this study, we tried to identify factors that hinder help seeking behavior by sorting only general adults who are currently working.

- The above-mentioned facts have been added to the introduction within a context. (Line 83-95)

11. Measures chapter should be revised extensively! There are hardly any citations. No item examples, scale (Likert? What range…?), reliability scores (Cornbach’s Alpha for example). Are any composite cores calculated. Also, was help-seeking behaviour assessed as a Yes/No answer? Specifically for seeking help because of depressive symptoms or in general? This chapter is lacking. Rational is missing for: Why were ages grouped like this? Why was income grouped like this (is it usually done like this in other articles)? Why was occupation grouped like this? – also why not unemployment (still wondering about this). 

- Response 11: Thank you for your comment. We have now modified the measure section as per your advice. (Measure section - marked in red)

- It was classified into four age type as per the Korean society, and the classification criteria were referred to previous studies using KCHS data. Relevant reference was also cited. (Line 138)

12. Nine points on the Patient Health Questionnaire-9 (PHQ-9). Why 9? It is just stated that you reviewed the literature, yet no argument is given. Is it the most common cut-off in Korean literature, what does it signify? I myself know of 5, 8 and 10 as usual cut-offs.

- Response 12: Thank you for your comment. We also know that many existing literatures present 10 points as cut-offs of PHQ-9. However, according to the studies subjected to Korean people, An et al (2013) suggested the optimal cut-off point was calculated as 9 points with 88.5% sensitivity and 94.7% specificity. According to Lim et al(2009), both 9 points and 10 points were found to have 88% sensitivity, but the specificity of 9 points was 86% and the specificity of 10 points was 52%.

- As your comment, we have now revised the text to explain why we used 9 points as cut-off. (Line 150-153, marked in yellow)

13. Subjective stress → please explain whether this has been done elsewhere or not? Is it original?

- Response 13: Thank you for your comment. Additional information on subjective stress measurement is described in Measure section. (Line154-157)

14. Furthermore, we conducted a factor analysis on the PHQ-9 to explore which depressive symptoms had a greater effect on help-seeking. → This information belongs in the Statistic Analysis chapter

- Response 14: Thank you for your suggestion. We have now revised the manuscript. (Statistical analysis section, marked in red, Line167)

15. Statistical Analysis chapter is also lacking in information. Which test were used exactly? Was checked for test prerequisites? I suggest building the sentences as follows: To analyse if … we conducted … tests after checking for … prerequisites (or in a similar manner). Remember. I have to understand exactly what you did and why so that (especially since you allow for data sharing) I could replicate your findings exactly after having read your article.

- Response 15: Thank you for your comment. In the case of factor analysis, the KMO measure of sampling adequacy was 0.874, and the approximate chi square value of Bartlett's test of sphericity was 2374168.654, and the significance (p-value) was <.001. Thus, it was confirmed as a suitable sample for factor analysis. This explanation is described in the main text. (Line 191)

- In the case of logistic regression analysis, even if the dependent variable does not satisfy the multivariate normality and assumption of homogeneity of variance required in discriminant function analysis, the dependent variable is It is known that it can be used if it corresponds to a binomial variable. Furthermore, it was confirmed that multicollinearity does not exist within the independent variable as shown below.

- 

16. Principal component analysis with direct oblimin rotation (δ=0) was used to extract the main factors with eigenvalues >1.0. → This is a good example for information that sadly doesn’t tell me why you did this?

- Response 16: Thank you for your comment. Since we expected a significant correlation between sub-factors of PHQ-9, principal component analysis with direct oblimin rotation (δ=0) was used to extract the main factors with eigenvalues >1.0. (Line 174)

17. Formatting: Use “.” for big numbers to approve readability. Sometimes you do this, sometimes not? Ex., this is annoying to read: Of the 561,654 people, 16,390 received more than nine points on the PHQ-9, 25,790 felt sad or hopeless

- Response 17: Thank you for your comment. For readability, it was indicated with %, and this information was transferred to 'data and subject' in the Method section and described. (Line 113-119, marked in blue)

18. Table 1 is a good overview. However, please format (e.g., capital letters someplace, other places small letter) and you must ad a “note” under the table to explain it. Tests used, “*”-meanings etc. (refer to reporting standards). Tables should be able to “stand by themselves” with no further information needed from the text (which is lacking due to the brief measures section anyhow).

- Response 18: Thank you for your comment. We modified table1 as per your advice. (Table1, marked in red)

19. A previous study (Jeon et al., 2020) using the BDI-II suggested that there were two latent factors of depression, and we also expected that the PHQ-9 would have two factors. As expected, two factors were extracted: one factor was related to somatic affective symptoms (items 1 to 5), and the other was a cognitive factor related to depression (items 6 to 9). → This belongs in the theory and methods chapter (at the very least the method chapter). Now I understand why you write this in the statistical analysis chapter. Please revise. (see comment #16)

- Response 19: Thank you for your comment. As per your comments, we moved this part from the results to the statistical analysis section. (Line 167-171)

20. Table 2 title is missing – please amend. The “note” information is lacking. See comment 18. Also, the note that is written is confusing, because no major loadings are shown in “bold face”?

- Response 20: Thank you for your comment. We have modified Table1 as per your advice. (Table2)

21. Results: Generally, why do the authors report who is less likely to seek help instead of saying who is more likely to do so. Is more positive and maybe also more helpful. At least it should always be juxtaposed to the reference groups. Ex.: Males were more likely 177 not to seek help for depression than females (aOR=1.31; 95% CI=1.157-1.487). Individuals aged 45-64 or 75 or more were more likely not to seek help for depression than those aged 19-44 (45-64 aged aOR=1.19; 95% CI=1.022-1.389; 75 or more aOR=1.45; 95% CI=1.059-1.973). → Possibly it is also a language issue: “more likely not to seek hep” = “less likely to seek help”

- Response 21: Thank you for your comment. That text has been corrected throughout the result section. Moreover, English proofreading of the entire manuscript was performed once more. (Result section, marked in red)

22. Table 3 better than 1 an 2. However, why are some aOR bold faced and others are not? The heading should also be more descriptive so that the reader knows straight away what they are looking at. O.R. should be aOR, otherwise not correct abbreviation → these mistakes should be revised before publication!

- Response 22: Thank you for your detailed comment. Table 3 is revised as per your recommendation. (Table 3)

23. For example, in the case of males, masculine social norms such as self-reliance and emotional control can make it difficult for males to seek help when they acknowledge their depression [27]. → The citation might not be chosen so well, because as I understand it refers to more “Western” masculinity norms; are there better sources for Korean men/masculinity norms? The same goes for the citation #28

- Response 23: Thank you for your comment. According to your advice, Korean research results have been added to the discussion. (Line 233)

24. The issue of socioeconomic status as a “fundamental cause” for health inequality should be discussed. Especially in light of these findings! The authors have great basis for building the SES-Index with the questions they assessed. It is not necessary, however, it might add positively to the regression model if the SES-Index is used. Irrespectively if done so or not, it should be discussed, especially as it was “teased at” as an obstacle in line 75)

- Response 24: Thank you for your comment. We have tried to interpret our findings in terms of health inequity, as per your advice. (Line 259-265)

25. The discussion is lacking and too short. I would expect more detail on the design and measurements and more detailed reflexion on the current research and how your findings fit into this “bigger picture”. Must be enhanced before publication.

- Response 25: Thank you for your comment. We have tried to give an in-depth interpretation by adding more parts to be discussed in the discussion section. Your advice has been very helpful. (Discussion section, marked in red, see also answer of no.24)

26. Generally: Some small English mistakes or uncommon use of words or interesting sentence structure – please revise for better readability. (ex.: Since approximately 230,000 participants completed the KCHS annually, a total of 915,089 participants completed it over four years, line 102-103)

- Response 26: Thank you for your comment. The English proofreading of the entire manuscript was performed once more for readability.

27. Yes - all data are fully available without restriction → How exactly? For non Korean readers it is impossible to understand the website. Is there a more internationally friendly option?

- Response 27: Thank you for your comment. Raw data for the Community Health Survey will be provided through the Community Health Survey website(https://chs.kdca.go.kr), and it can be downloaded after applying through a request for raw data. However, the English version homepage is not provided.

Reviewer #2

Background

1. Please provide information on help for depression available in Korea. How easy or costly is it to seek help for depression?

- Response 1: Thank you. Korea has relatively low cost barriers for medical treatment. These explanations have been presented to the introduction section. (Line77-82)

Methods 

2. How many people participated in KCHS in total – what percentage makes 6505?

- Response 2: Thank you for your comment. It has been rewritten to make it easier to read about the number and proportion of participants. (Line 113-120, marked in blue)

3. Please present the inclusion and exclusion criteria more clearly. Why were the exclusion criteria chosen?

- Response 3: Thank you for your comment. The inclusion criteria of KCHS data and related reference was cited (Line 108), and the exclusion criteria implemented for use in this study was described. (Line 113-115, marked in red)

4. What have been the prevalence for depression in Korea in the time 2017 to 2020? Are the people identified in your investigated sample representative for the prevalence?

- Response 4: Thank you. The depression experience rate of total subjects was 5.0~5.8% from 2017 to 2020, and that of the sample used for our analysis was 4.59%. We have added this in result section. (Line 180)

5. Was help seeking assessed by self-report on one item, multiple items or validated scales? Please specify the assessment. (ll 111-113)

- Response 5: Thank you for your comment. Although it is a single item, it is judged to be highly reliable because it is data confirmed through direct interviews rather than self-report questionnaires. As you recommended, we have added this information to the measure section. (Line 123-125)

Results

6. Ll151 -154: Please specify which populations you investigated. There is obviously the total population of: 915,089, then there is people with depressive symptoms: 9505, then there is people seeking help in the total population 561,654 .. It is quite confusing with all the different populations to understand, which populations your statistics were based on. Please find a way to make it easier to follow. Please also present the relative frequencies in regard to the total population of 915,089.

- Response 6: Thank you for your comment. It has been rewritten to make it easier to read about the number and proportion of participants along with the responses to comment #2. (Line 113-120, marked in blue)

7. Ll 154: I understand that 561,654 people sought help – but then of the depressed people only 1781 were seeking help. I thought, you filtered for help seeking in the beginning? What is the difference between help seeking and professional counseling? Please also provide these data in table 1.

- Response 7: Thank you for your comment. The total working population, 561,654 are the participants of this study. Of the 561,654, 6,505 were screened for depression through PHQ-9 and it was confirmed that they were suffering from depression in their daily life. Of the 6,506, 1,781 were those who sought help from professional with depression. (Line 113-120)

- In this study, help-seeking behavior was measured according to whether they had ever received expert advice for depression (visiting a medical institution, professional counseling institution, or public health center). (Line 126-131)

- As per your advice, we have added these data to Table1. (Line 187)

8. What population was the factor analysis based on? Please provide N?

- Response 8: Thank you for your question. Factor analysis for PHQ-9 was performed on all participants (N=915,089) who participated in the KCHS from 2017 to 2020. N was presented in the title of table. (Line 196)

9. Please explain what aOR is? Please incorporate that abbreviation in the text.

- Response 9: Thank you for your comment. The first time use an abbreviation in the text, we presented both the spelled-out version and the short form. (Line 204)

10. In table 3 you present OR – or is this also aOR?

- Response 10: Thank you for your comment. The Adjusted OR is correct. It was incorrectly written as OR, so it was corrected to adjusted OR. (Table 3)

Discussion

11. Please discuss the following limitations: PHQ-9 just a screener for depressive symptoms, there was no psychological interview validating a depression diagnosis. Please also discuss potential biases of non-reporting for help seeking or recall bias.

- Response 11: Thank you for your comment. We agree with you. As per your recommendations, we have now added the limitations to the manuscript. (Line 272-276)

Tables

12. Please provide a legend below the table 1 explaining abbreviations and asterisks

- Response 12: Thank you for your comment. We have now added a legend below the Table 1. (Table1)

---

## [Decision Letter · Decision Letter 1]

15 Nov 2022

PONE-D-22-18284R1The Effects of Sociodemographic Factors on Help-Seeking for Depression Based on the 2017-2020 Korean Community Health SurveyPLOS ONE

Dear Dr. Yun,

Thank you for submitting your manuscript to PLOS ONE. After careful consideration, we feel that it has merit but does not fully meet PLOS ONE’s publication criteria as it currently stands. Therefore, we invite you to submit a revised version of the manuscript that addresses the points raised during the review process.

There are still some issues related to coding and limitations to be sorted out. Both reviewers also strongly suggest the paper be revised by a professional service. Please submit your revised manuscript by Dec 30 2022 11:59PM. If you will need more time than this to complete your revisions, please reply to this message or contact the journal office at plosone@plos.org. Please include the following items when submitting your revised manuscript:A rebuttal letter that responds to each point raised by the academic editor and reviewer(s). You should upload this letter as a separate file labeled 'Response to Reviewers'.A marked-up copy of your manuscript that highlights changes made to the original version. You should upload this as a separate file labeled 'Revised Manuscript with Track Changes'.An unmarked version of your revised paper without tracked changes. You should upload this as a separate file labeled 'Manuscript'.If applicable, we recommend that you deposit your laboratory protocols in protocols.io to enhance the reproducibility of your results. Protocols.io assigns your protocol its own identifier (DOI) so that it can be cited independently in the future. For instructions see: https://journals.plos.org/plosone/s/submission-guidelines#loc-laboratory-protocols. Additionally, PLOS ONE offers an option for publishing peer-reviewed Lab Protocol articles, which describe protocols hosted on protocols.io. Read more information on sharing protocols at https://plos.org/protocols?utm_medium=editorial-email&utm_source=authorletters&utm_campaign=protocols.

We look forward to receiving your revised manuscript.

Kind regards,

Pedro Vieira da Silva Magalhaes, M.D., Ph.D.

Academic Editor

PLOS ONE

Journal Requirements:

Reviewers' comments:

Reviewer's Responses to Questions

**Comments to the Author**

1. If the authors have adequately addressed your comments raised in a previous round of review and you feel that this manuscript is now acceptable for publication, you may indicate that here to bypass the “Comments to the Author” section, enter your conflict of interest statement in the “Confidential to Editor” section, and submit your "Accept" recommendation.

Reviewer #1: (No Response)

Reviewer #2: All comments have been addressed

2. Is the manuscript technically sound, and do the data support the conclusions?

Reviewer #1: Yes

Reviewer #2: Yes

3. Has the statistical analysis been performed appropriately and rigorously? 

Reviewer #1: Yes

Reviewer #2: Yes

4. Have the authors made all data underlying the findings in their manuscript fully available?

Reviewer #1: Yes

Reviewer #2: Yes

5. Is the manuscript presented in an intelligible fashion and written in standard English?

Reviewer #1: No

Reviewer #2: Yes

6. Review Comments to the Author

Reviewer #1: Dear authors,

Thank you for incorporating some of my criticism into your work. I have reread the article and the points you changed and answer in detail in the table below. I have crossed through the points that I find have been satisfactorily changed. Some points should however still be considered (my new comments in red).

I know you are not native English speakers so I kindly suggest you let a professional proof reader correct your draft.

Yours sincerely and all the best in your work,

Your anonymous reviewer

(see attached file for comments)

Reviewer #2: It is highly recommended that the manuscript is edited by a native speaker. Some phrases and paragraphs sound a bit off and could benefit from a language check.

7. PLOS authors have the option to publish the peer review history of their article (what does this mean?). If published, this will include your full peer review and any attached files.

Reviewer #1: No

Reviewer #2: No

---

## [Author Response · Author response to Decision Letter 1]

29 Dec 2022

ANSWERS TO REVIEWERS

We thank the reviewer for your thoughtful suggestions and insights. The manuscript has benefited from these insightful suggestions. The manuscript has been rechecked and the necessary changes have been made in accordance with the reviewers’ suggestions. The responses to all comments have been prepared and attached herewith.

1. Please write in the abstract more precisely. For example: “(…) that males, aged 45-64 years and those more than 75 years, who did not have Medical Aid program, and less educated people were less likely to seek help for depression.” → This sentence is too ambiguous. This is only one example that the authors find it difficult to use clear English. The article would benefit from a professional proof reader.

Thank you for your advice. We have revised the abstract accordingly. Our manuscript has also been proofread thoroughly. (Line 40-44)

Male participants (adjusted odds ratio [aOR]=1.31, 95% confidence interval [CI]=1.157–1.487), those aged 45–64 years (aOR=1.192, 95% CI=1.022–1.389) and more than 75 years (aOR=1.446, 95% CI=1.059–1.973), those not having a Medical Aid program (aOR=1.750, 95% CI=1.375–2.226), and those having low educational levels (aOR=.896, 95% CI=.830–.968) were less likely to seek professional help for depression.

2. Although I appreciate the changes made, I still find the exclusion to be problematic. It would, in my opinion, be more prudent to keep people with no current employment in the sample and categorise them as the 5. category in “occupation”. In the way the rational is presented here, there is an implicit discrimination in the study against people with no current employment. You argue “Since various sociodemographic factors are related to help-seeking behavior and depression, it is difficult to generalize results of research to all population groups in the community”(ll. 84-86) and then exclude a part of the population to analyse the “productive population”, making the findings in this study less generalizable.

Thank you for your comment. Previous studies on help-seeking behavior have focused on vulnerable groups such as older adults and individuals with low socioeconomic status. Therefore, our study aimed to conduct an analysis targeting the working population to differentiate from the existing research topics. In addition, being unemployed can be a reason or a consequence of depression. However, as you indicated, if we had controlled for this factor in the analysis, it may have been more representative. We have added these points in the limitations section. (Lines 269-271)

Second, this study targeted employed individuals alone; therefore, bias cannot be ruled out, and the results of this study cannot represent the entire community population.

3. All in all this is a lot better! However, reliability scores are still missing for PHQ-9 and subjective stress (why?)

Thank you for your comment. We have analyzed the reliability score and added it. (Line 155)

Cronbach's alpha for the PHQ-9 was 0.802 in the present study.

4. This is still not satisfactorily explained (or maybe the KCHS wrote the items themselves without reference to previous studies?)

The KCHS survey gathered information through face-to-face interactions between trained interviewers and respondents. Data on subjective stress were also obtained through interviews using one simple question. We have added these points in the limitations section. (Lines 274-277)

although a trained interviewer conducted the interview to measure depression and subjective stress, the participants were not clinically diagnosed with depression by a mental health professional. Moreover, subjective stress was measured by one simple question rather than a validated tool.

5. The changes are helpful, but why not use . instead of , to write big numbers?

Thank you for your comment. In the US, UK, and China, a comma is placed after every 3 digit positions for numbers larger than 999. We have followed the guidelines of this journal.

6. Table 2 title is still missing. These superficial things should be done with care! Please amend.

Thank you for your comment. We have added the table title as per your advice. (Line 197)Table2. The factor analyses of the PHQ-9 in the sample at study entry (N=915,089)

7. Generally in multiple regressions, it is more usual to report the findings in reference to the reference group. I didn’t notice this before due to the double negative, but for example I suggest writing the results as follows: “It was found X were more likely to seek help for depression than Y” (and Y is the reference group)

Also, there is a confusing presentation of the multiple regression results that need to be simplified. You code “help-seeking” as 0 and “non-help seeking” as 1. That would mean that those groups with aOR>1 were more likely not to seek help compared to the reference group and this leads to confusing double negatives in the results section: e.g.: “Individuals who did not have Medical Aid program were less likely to seek help for depression than individuals with Medical Aid program” (l. 205). It would be a lot easier for the reader if a) the help-seeking outcome were coded as 0=no help and 1=help seeking and the results section written with simple “X was more likely then Y to seek help” or “X was less likely then Y to seek help” (Y being the reference category).

I could only find the categorisation as 0=help seeking and 1=no help-seeking in the note of table 3, which I first didn’t read and then was confused to the result interpretation. It should also be transparently written so in the paragraph “statistical analysis”.

Thank you for your comment. Our research goal was to identify factors related to help-seeking behavior to guide appropriate interventions. To emphasize the subject of this study, no help-seeking behavior was coded as '1' and analyses were performed. Based on your suggestion, we have added an explanation about the coding in the Method section to help readers understand better. (Lines 165-166)

Help-seeking behavior was coded as 0, which stands for help-seeking behavior and 1, stands for no help-seeking behavior.

8. English is still lacking in clarity. And some sentences were not easy to understand.

Thank you for your comment. The entire manuscript has been proofread thoroughly.

---

## [Editor Report · Decision Letter 2]

5 Jan 2023

The Effects of Sociodemographic Factors on Help-Seeking for Depression Based on the 2017-2020 Korean Community Health Survey

PONE-D-22-18284R2

Dear Dr. Yun,

We’re pleased to inform you that your manuscript has been judged scientifically suitable for publication and will be formally accepted for publication once it meets all outstanding technical requirements.

Kind regards,

Pedro Vieira da Silva Magalhaes, M.D., Ph.D.

Academic Editor

PLOS ONE
---

## [Editor Report · Acceptance letter]

9 Jan 2023

PONE-D-22-18284R2 

The Effects of Sociodemographic Factors on Help-Seeking for Depression:
Based on the 2017–2020 Korean Community Health Survey 

Dear Dr. Yun:

I'm pleased to inform you that your manuscript has been deemed suitable for publication in PLOS ONE. Congratulations! Your manuscript is now with our production department. 

Kind regards, 

on behalf of

Professor Pedro Vieira da Silva Magalhaes 

Academic Editor

PLOS ONE